# The Antiasthma Medication Ciclesonide Suppresses Breast Cancer Stem Cells through Inhibition of the Glucocorticoid Receptor Signaling-Dependent YAP Pathway

**DOI:** 10.3390/molecules25246028

**Published:** 2020-12-19

**Authors:** Su-Lim Kim, Hack Sun Choi, Ji-Hyang Kim, Dong-Sun Lee

**Affiliations:** 1Interdisciplinary Graduate Program in Advanced Convergence Technology & Science, Jeju National University, Jeju 63243, Korea; ksl1101@naver.com (S.-L.K.); choix074@jejunu.ac.kr (H.S.C.); seogwil@naver.com (J.-H.K.); 2Practical Translational Research Center, Jeju National University, Jeju 63243, Korea; 3Bio-Health Materials Core-Facility Center, Jeju National University, Jeju 63243, Korea; 4Faculty of Biotechnology, College of Applied Life Sciences, Jeju National University, SARI, Jeju 63243, Korea; 5Subtropical/Tropical Organism Gene Bank, Jeju National University, Jeju 63243, Korea

**Keywords:** CSCs, ciclesonide, glucocorticoid receptor, YAP, glucocorticoid

## Abstract

Ciclesonide is an FDA-approved glucocorticoid used to treat asthma and allergic rhinitis. However, whether it has anticancer and anti-cancer stem cell (CSC) effects is unknown. This study focused on investigating the effect of ciclesonide on breast cancer and CSCs and determining its underlying mechanism. Here, we showed that ciclesonide inhibits breast cancer and CSC formation. Similar glucocorticoids—dexamethasone and prednisone—did not inhibit CSC formation. Ciclesonide-induced glucocorticoid receptor (GR) degradation was dependent on ubiquitination. We showed via GR small interfering RNA (siRNA) that GR plays an important role in CSC formation. We showed via western blot and immunofluorescence assays that ciclesonide reduces the nuclear level of GR. The GR antagonist RU-486 also inhibited CSC formation. Ciclesonide reduced the protein level of the Hippo transducer Yes-associated protein (YAP). GR siRNA induced a decrease in YAP protein expression and inhibited mammosphere formation. The YAP inhibitor verteporfin inhibited CSC formation and transcription of the connective tissue growth factor and cysteine-rich protein 61 genes. The GR/YAP1 pathway regulated breast CSC formation. We showed that the GR/YAP signaling pathway regulates breast CSC formation and revealed a new approach for targeting GR and YAP to inhibit CSC formation.

## 1. Introduction

The Hippo pathway, discovered in the fruit fly *Drosophila melanogaster*, is a tumor suppressor signaling pathway that controls cell growth, tissue homeostasis, and organ size through the regulation of cell proliferation and apoptosis [1]. The main kinases in the Hippo signaling pathway are the serine/threonine kinase Mst1/2 and the tumor suppressor LATS1/2 [2]. The LATS1/2 kinase phosphorylates and inhibits the Hippo pathway effectors Yes-associated protein (YAP) and Tafazzin (TAZ). Phosphorylated YAP/TAZ was degraded in cytosol and dephosphorylated YAP/TAZ are localized in the nucleus [3]. YAP/TAZ, transcriptional coactivators, and Transcriptional enhancer factor (TEA) domain family member (TEAD), a transcription factor complex, induce the expression of genes mediating cell growth, proliferation, and survival [3,4]. Activation of YAP/TAZ has been found in human cancers, including metastatic breast cancer, and increases the self-renewal of cancer stem cells (CSCs) [4,5,6]. Aberrant activation of YAP/TAZ induces CSC traits, anoikis resistance, epithelial-mesenchymal transition (EMT), drug resistance, and metastasis [2]. Steroid hormones regulate a physiological process through their binding to transcription factors, the estrogen receptor (ER), the progesterone receptor (PR), and the glucocorticoid receptor (GR) [7]. GR is divided into five splice variants, GR-α, GR-β, GR-γ, GR-P, and GR-A. The GR-α isoform is responsible for glucocorticoid (GC)-mediated transcriptional activity. GR is a mediator of GCs related to stress, and stress contributes to cancer progression [8]. The activation of GR is fundamental for CSCs self-renewal and chemoresistance of breast cancer cells, and GR signaling affects the mechanical properties of the tumor microenvironment, ultimately promoting YAP nuclear accumulation and activation [1]. GCs control apoptosis and proliferation in breast cancer models [9]. An increase in stress hormones during breast cancer progression results in GR activation at distant metastatic sites and reduces survival [10]. GR activation induces heterogeneity and metastasis, and GCs promote breast cancer metastasis. Doctors must be cautious in the use of GCs to treat breast cancer patients who have developed cancer-related complications [10]. Radiation-induced GR expression increases the CSC population in prostate cancer through SGK1-Wnt/β-catenin signaling, thus limiting the efficacy of radiotherapy for prostate cancer treatment [11].

The antiasthma medication ciclesonide is a GC used to treat asthma and allergic rhinitis. This new inhaled corticosteroid is effective as a once-daily controller therapy for pediatric asthma and reduces airway inflammation through once-daily administration [12]. Ciclesonide has strong anti-inflammatory activity in vitro and in vivo. The relative binding affinity of ciclesonide to rat GR is higher than that of dexamethasone (DEX) [13]. We showed that ciclesonide inhibited proliferation of breast cancer and CSC formation by suppressing the GR signaling-dependent YAP signaling pathway. The GR/YAP axis is thus a therapeutic target for controlling breast CSCs in breast cancer.

## 2. Results

### 2.1. Ciclesonide Induces Proliferation Inhibition and Apoptosis of Breast Cancer Cells

We tested the inhibitory effect of ciclesonide on the proliferation of breast cancers and found that ciclesonide showed anti-proliferative effects (Figure 1A,B). Ciclesonide induced the apoptosis of breast cancer cells (Figure 1C). Ciclesonide increased caspase 3/7 activity in a caspase concentration-dependent manner (Figure 1D). After treatment with ciclesonide, breast cancer cells exhibited apoptotic body formation (Figure 1E). In addition, ciclesonide inhibited migration and colony formation (Figure 1F,G). Our results show that ciclesonide effectively suppresses the cancer hallmarks of cell migration, proliferation, and colony formation of breast cancer.

### 2.2. Ciclesonide Inhibits Tumor Growth

As ciclesonide has an anti-proliferative effect on breast cancer, we used an in vivo mouse model to examine whether it reduces tumor growth. The body weights of control and ciclesonide-treated mice did not change (Figure 2A). The weights of tumors from ciclesonide-treated nude mice were lower than those of tumors from control nude mice (Figure 2B). The volumes of tumors from ciclesonide-treated mice were smaller than those of tumors from control mice (Figure 2C). Our results indicated that ciclesonide effectively reduced tumor growth.

### 2.3. Effect of Ciclesonide, Prednisone, and Dexamethasone on Breast CSCs

To test whether ciclesonide inhibits the formation of mammospheres derived from MDA-MB-231 and MCF-7 cells, we treated mammospheres with different concentrations of ciclesonide. Ciclesonide inhibited breast mammosphere formation (Figure 3A and Appendix A). We assessed the mammosphere formation using the most commonly used steroid hormones, prednisone and dexamethasone. Neither of these two steroids affected mammosphere formation (Figure 3B). The breast CSC population is characterized by CD44^+^/CD24^−^ phenotype. To examine the inhibitory effect of ciclesonide on CD44^+^/CD24^−^ population, the CD44^+^/CD24^−^ population of breast cancer cells was assayed under ciclesonide treatment. Ciclesonide reduced the CD44^+^/CD24^−^ cell proportion from 13.6% to 3.0% (Figure 3C). As aldehyde dehydrogenase (ALDH) is marker of cancer stem cells, we also examined the inhibitory effect of ciclesonide on ALDH-positive cancer cells. Ciclesonide decreased the ALDH-positive cell proportion from 1.0% to 0.4% (Figure 3D). Our results indicate that ciclesonide specifically inhibits mammosphere formation.

### 2.4. Ciclesonide Attenuates Mammosphere Formation via GR Inhibition

To determine the cellular mechanism of ciclesonide in mammosphere formation, the level of GR protein was examined in mammospheres, as ciclesonide binds GR. GR levels were reduced significantly under ciclesonide treatment but not dexamethasone treatment (Figure 4A). However, the transcription level of the GR gene was not changed under treatment with either ciclesonide or dexamethasone (Figure 4B). Treatment with the proteasome inhibitor MG132 protected GR from ciclesonide-induced degradation, suggesting that ciclesonide increased the proteasomal degradation of GR (Figure 4C). We performed immunoprecipitation with an anti-ubiquitin antibody and western blotting with an anti-GR antibody in ciclesonide-treated cells treated. The level of ubiquitinated GR was increased under ciclesonide treatment (Figure 4D). These data suggest that ciclesonide induces the degradation of the GR protein via an ubiquitin-dependent pathway (Figure 4C,D).

### 2.5. The GR Signaling Pathway Is an Important Signaling Pathway for Breast CSC Formation

To determine whether GR is essential for CSCs, we examined CSC formation in response to siRNA-mediated silencing of GR. Breast cancer cells with siRNA-mediated specific knockdown of GR showed a 50% reduction in tumorsphere formation (Figure 4E). Our data showed that nuclear GR levels were significantly reduced under ciclesonide treatment (Figure 4F), and our immunofluorescence data showed that ciclesonide-treated cells showed lower levels of nuclear GR than untreated cancer cells (Figure 4G). Treatment with RU486, a GR antagonist, inhibited the formation of mammospheres (Figure 4H). Our observations suggest that GR and GR signaling are essential for mammosphere formation.

### 2.6. Ciclesonide Regulates the GR/YAP Signaling Axis

We examined whether ciclesonide regulates the GR/YAP signaling axis. YAP levels were significantly reduced under ciclesonide treatment. In addition, the transcription level of the YAP gene was changed under ciclesonide treatment (Figure 5A). The nuclear YAP level was significantly reduced under ciclesonide treatment (Figure 5B). Cancer cells with YAP downregulation via YAP-specific siRNA showed a 50% reduction in mammosphere formation (Figure 5D). Cells with GR downregulation via GR-specific siRNA showed a reduction in the YAP protein level (Figure 5C). To examine whether ciclesonide regulates the YAP signaling axis, we used verteporfin, a YAP inhibitor that inhibits the physical YAP-TEAD interaction [14]. Verteporfin dramatically reduced breast CSC formation (Figure 5E). We examined the transcription levels of YAP target genes (*Ctgf* and *Cyr65*) in mammospheres derived from breast cancer cells and found that the mRNA levels of CTGF and CYR61 were reduced under verteporfin treatment. We examined the correlation between GR (*NR3C1*) and YAP gene expression in breast cancer patients from a publicly available data from The Cancer Genome Atlas (TCGA). YAP gene expression showed a significant positive correlation with GR (*NR3C1*) expression in breast invasive carcinoma (Figure 5G). These results show that ciclesonide regulates the GR/YAP1 signaling axis to inhibit breast CSC formation.

### 2.7. Ciclesonide Inhibits the Expression of Cancer Stem Cell Marker Genes and Mammosphere Growth

To examine whether ciclesonide inhibits CSC marker genes, we evaluated the transcriptional levels of CSC marker genes. Ciclesonide inhibited gene transcriptions such as sex determining region Y (SRY)-box 2 (Sox2), Nanog, octamer-binding transcription factor 4 (Oct4), and CD44 in breast CSCs (Figure 6A). To examine whether ciclesonide decreases mammosphere growth, we added ciclesonide to the culture medium and counted the cancer cells derived from the mammospheres. Ciclesonide induced mammosphere cell death (Figure 6B). These data indicate that ciclesonide induces a reduction in mammosphere growth and inhibits breast CSC formation through deregulation of the GR/YAP1 signaling axis (Figure 6C).

## 3. Materials and Methods

### 3.1. Cell and Mammosphere Culture

Two breast cancer cell lines, MCF-7 and MDA-MB-231, were obtained from the American Type Culture Collection (Rockville, MD, USA). All human breast cancer cells were maintained in DMEM supplemented with 10% fetal bovine serum (FBS) (ThermoFisher Scientific, Waltham, MA, USA) and 1% penicillin/streptomycin (HyClone, ThermoFisher Scientific). To establish primary mammospheres, single-cell suspensions of MCF-7 and MDA-MB-231 cells were seeded at a density of 3.5 × 10^4^–4 × 10^4^ and 0.5 × 10^4^–1 × 10^4^ cells/well, respectively, in ultralow attachment 6-well plates containing 2 mL of complete MammoCult^TM^ medium (StemCell Technologies; Vancouver, BC, Canada), which was supplemented with 4 μg/mL heparin, 0.48 μg/mL hydrocortisone, 100 U/mL penicillin, and 100 μg/mL streptomycin. All cells were maintained in a humidified 5% CO_2_ incubator at 37 °C for 7 days. After 7 days of culture, 8-bit grayscale images of the mammospheres were acquired by placing the cell culture plate on a scanner (Epson Perfection V700 PHOTO, Epson, Tokyo, Japan). Low-resolution images (300 dpi) were loaded using the software program NICE (ftp://ftp.nist.gov/pub/physics/mlclarke/NICE) [15]. For counting, regions of interest (ROIs) were created by choosing the desired number of rows and columns (e.g., 2 × 3 for a 6-well plate), and individual ROIs were defined by moving and resizing the provided ROI shapes after selecting the elliptical setting in the NICE program. The background signal of the images was negated using thresholding algorithms, and the selected images were automatically counted. The results of the mammosphere formation assay are reported as the mammosphere formation efficiency (MFE, % of control), which corresponds to the number of drug-treated mammospheres per well/the number of control mammospheres per well ×100 as previously described [16].

### 3.2. Antibodies and Small Interfering RNAs (siRNAs)

Anti-YAP, anti-pYAP, and anti-GR antibodies were purchased from Cell Signaling Technology (Danvers, MA, USA). Anti-ubiquitin, anti-β-actin, and anti-Lamin b antibodies were purchased from Santa Cruz Biotechnology (Dallas, TX, USA). Anti-CD44 FITC and anti-CD24 PE antibodies were obtained from BD (San Jose, CA, USA). Human GR- and YAP-specific siRNAs were obtained from Bioneer COR (Daejeon, Korea).

### 3.3. Cell Proliferation

We followed a previous method [17]. The MTS assay was conducted as the cell viability assay in this study. Breast cancer cells (10^4^ cells/well) were cultured in a 96-well plate for 24 h. The breast cancer cell lines were treated with increasing concentrations of ciclesonide (0, 5, 10, 20, 40, and 80 µM) for 1 day. Cell proliferation was assessed using a CellTiter 96^®^ Aqueous One Solution cell kit (Promega, Madison, WI, USA). After mixing DMEM and aqueous one solution (5:1), we added 100 µL of mixture to each well and incubated the cells at 37 °C for 1 h and the OD_490_ was measured using a microplate reader (SpectraMax, San Jose, CA, USA).

### 3.4. Colony Formation and Migration Assays

For the colony formation assay, MDA-MB-231 cancer cells (1000 cells/well) were incubated with ciclesonide for 7 days in DMEM containing 10% FBS and 1% penicillin/streptomycin. The colonies were washed three times with 1X PBS, fixed for 10 min using 4% formaldehyde, and stained for 1 h with 0.04% crystal violet. After washing twice with distilled water, we acquired images using a scanner (Epson Perfection, Epson, Tokyo, Japan). The colonies were counted with the NICE software program [15]. For the migration assay, MDA-MB-231 cells were cultured in a 6-well plate with 2 × 10^6^ cells/plate. After 24 h, the cells were scratched using a microtip. The cells were washed two times with 1× PBS and cultured with ciclesonide in DMEM for 8 h and 14 h. Photographs of the migrated areas were acquired using a light microscope [18].

### 3.5. Annexin V/PI Staining Assay and Analysis of Cell Apoptosis

For the Annexin V/PI assay, breast cancer cells were incubated with ciclesonide (20 μM) in 6-well plates. Apoptotic cells were analyzed by FITC-Annexin V/PI staining method according to the manufacturer’s instructions (BD, San Jose, CA, USA). The stained samples were detected by Accuri C6 (BD, San Jose, CA, USA). For Hoechst 33258 staining, cancer cells were incubated with 20 μM ciclesonide for 24 h, and were then incubated with Hoechst 33258 (10 mg/mL) solution for 30 min at 37 °C. The cells were observed under a fluorescence microscope (Lionheart FX, BioTek, Winooski, VT, USA).

### 3.6. Flow Cytometric Analysis and Aldehyde Dehydrogenase (ALDH1) Activity Assay

We used a previously described method [18]. Cancer cells were detached by using 1X trypsin/EDTA. The detached cells were washed with Fluorescence-activated Cell Sorting (FACS) buffer and suspended with 100 µL of FACS buffer. We added 10 µL of FITC-conjugated anti-human CD44 and PE-conjugated anti-human CD24 to each sample. The samples were incubated on ice for 20 min, washed two times with 1X FACS buffer, and the isolated cells were then centrifuged and washed two times with 1× FACS buffer. The cell pellet was analyzed using an Accuri C6 flow cytometer (BD, San Jose, CA, USA). ALDH1 activity was assessed using an ALDEFUOR^TM^ assay kit (STEMCELL Technologies) via a previously described method [18]. Cancer cells were incubated in ALDH assay buffer at 37 °C for 30 min. ALDH-positive cells were examined by using an Accuri C6 (BD, San Jose, CA, USA).

### 3.7. Gene Expression Analysis

Total RNA was isolated using MDA-MB-231 cells. RT-qPCR was performed using a real-time One-Step RT-qPCR kit (Enzynomics, Daejeon, Korea). We followed a previously described method, and we made RT-qPCR mixture containing TOPrealTM One-step RT qPCR Enzyme MIX 1 µL, 2X TOPrealTM One-step RT qPCR Reaction MIX (with low ROX) 10 µL, RNA template (100 ng/µL) 1µL, specific primers-F (10 ng/µL) 1 µL, specific primers-R (10 ng/µL) 1 µL, and sterile water 6 µL in each samples. The relative transcript expression levels of the target genes were analyzed using the comparative CT method [17]. The specific primers are described in Appendix A. PCR experiments were tested to allow statistical analysis. The β-actin gene was used as an internal control.

### 3.8. Western Blot Analysis

Protein extracts were isolated from breast cancer cells and mammospheres. We cultured MDA-MB-231 cells to ultra-low attachment 6-well plate for 5 days and treated 10 µM of ciclesonide for 2 days. We washed the cell pellet twice for 1× PBS. The cells were resuspended in buffer A (pH 7.9 of 10 mM Hepes, 1.5 mM MgCl2, 10 mM KCl, 0.05% NP-40, 0.5 mM DTT, 10 mM protease inhibitor, 10 mM NaF, and 10 mM Vanadate) and the lysate was microcentrifuged at 10,000× *g* for 5 min to pellet the nuclei. The supernatant contains cytosol fraction and the resulting pellet contains the nucleus. The nuclear pellet was dissolved with Radio-Immunoprecipitation Assay (RIPA) buffer with 10mM protease inhibitor, 10mM NaF, and 10mM Vanadate, and then pipetting. The samples were put on ice for 30 min and then microcentrifuged at 14,000× *g* for 15 min. The resulting supernatant contained the nuclei. After electrophoresis on 10% sodium dodecyl sulfate-polyacrylamide gel electrophoresis (SDS-PAGE) gels, proteins were transferred to polyvinylidene fluoride (PVDF) membranes (EMD Millipore, Burlington, MA, USA). Membranes were incubated first in Odyssey blocking buffer for 1 h and then overnight with primary antibodies. The anti-GR, anti-YAP, anti-pYAP, anti-lamin B, and anti-β-actin antibodies were used. After washing and drying, membranes were incubated with IRDye 680RD- and 800W-conjugated secondary antibodies, and images were acquired using an ODYSSEY CLx (LI-COR, Lincoln, NE, USA).

### 3.9. Caspase-3/7 Assay

We used a previously reported method [19]. Cancer cells were cultured with ciclesonide. Caspase-3/7 activity was measured using a Caspase-Glo 3/7 kit (Promega, Madison, WI, USA) according to the manufacturer’s instructions. Then, 100 μL of Caspase-Glo 3/7 reagent was added to 96-well plates and incubated, and the activity was measured in a GloMax^®^ Explorer luminometer (Promega, Madison, WI, USA).

### 3.10. siRNA

To examine the inhibitory effect of GR and YAP on mammosphere formation, we transfected cancer cells with siRNAs targeting human GR and YAP (Bioneer, Daejeon, Korea). The GR and YAP siRNAs (NM_181651.1) and a scrambled siRNA were obtained from Bioneer (Daejeon Cor., Korea). For siRNA transfection, we cultured MDA-MB-231 cells to be 70–90% confluent. After the cells were attached to plate, we diluted Lipofectamine^®^ 3000 reagent 4 µL in Opti-MEM^®^ medium 125 µL and prepared master mix of SiRNA by diluting SiRNA 5 µg in Opti-MEM^®^ medium 125 µL in each tube. Then, we mixed the diluted SiRNA and diluted Lipofectamine^®^ 3000 reagent (for control, only Opti-MEM^®^ and diluted Lipofectamine^®^ 3000 should be mixed with a scrambled siRNA) and incubated it for 5 min at room temperature. We added siRNA-lipid complex to each well and incubated cells for 2–4 days at 37 °C. The protein levels of GR and YAP were determined via western blot analysis.

### 3.11. Immunoprecipitation (IP)

Mammospheres were washed with 1× PBS and resuspended in lysis buffer. The lysates were incubated with an anti-ubiquitin antibody for 16 h at 4 °C. Protein A/G agarose (ThermoScientific, Waltham, MA, USA) was added to the mixtures. The mixtures were centrifuged, and the precipitates were washed with lysis buffer 5 times, run on SDS-PAGE gels, and subjected to immunoblotting.

### 3.12. Immunofluorescence (IF)

Breast cancer cells were fixed with 4% paraformaldehyde for 20 min, permeabilized with 0.5% Triton X-100 for 15 min, blocked with 3% bovine serum albumin (BSA) for 1 h, and labeled with a mouse anti-GR antibody followed by an Alexa Fluor 488-conjugated anti-mouse secondary antibody. We used nonspecific signal conditions to confirm the specificity of the primary antibodies for immunofluorescence. Finally, nuclei were stained with DAPI, and GR was visualized with a fluorescence microscope (Lionheart, BioTek, VT, USA).

### 3.13. Xenograft Transplantation

Twelve female nude mice were injected with two million MDA-MB-231 cells and injected with/without ciclesonide (10 mg/kg). Tumor volumes were estimated for 45 days using the following formula: (width^2^ × length)/2. The mouse experiments were performed as reported previously [20]. Animal care and experimentation were performed in accordance with protocols approved by the Institutional Animal Care and Use Committee (IACUC) of Jeju National University. Female nude mice (5 weeks old) were purchased from OrientBio (Seoul, South Korea) and maintained in a mouse facility for 1 week.

### 3.14. Statistical Analysis

All data were processed with GraphPad Prism 5.0 software (GraphPad Prism Inc., San Diego, CA, USA). All data values are reported as the means ± standard deviations (SDs). Data were analyzed with one-way ANOVA. *p*-values of less than 0.05 were considered to indicate significance.

## 4. Discussion

Although dramatic advances have been made in cancer research, breast cancer remains an important health problem. Breast cancer is the most common cancer affecting women, and recently, researchers have focused on young breast cancer patients aged <45 years [21]. This cancer is heterogeneous, complex, and aggressive. Breast CSCs are a key factor in tumor heterogeneity and cause chemoresistance and metastasis [22]. Breast CSCs can survive in target organs and generate metastasis up to two decades after diagnosis [22]. Biomarkers of breast CSCs, including CD44 and aldehyde dehydrogenase 1 (ALDH1), can be regulated during cancer progression and metastasis. In triple negative breast cancer patients, CD44 promotes the transcription of PD-L1, an immune checkpoint, through its cleaved intracytoplasmic domain (ICD). Inhibition of ALDH1 in breast cancer by curcumin decreased multidrug resistance.

In this report, we showed that ciclesonide inhibits breast cancer cell proliferation and CSC formation. Similar GCs—dexamethasone and prednisone—did not inhibit CSC formation (Figure 3 and Appendix A). Clinically, GCs such as dexamethasone have been used to treat cancer patients to reduce the side effects of chemotherapy, reduce nausea, and protect healthy tissue [23]. Clinical evidence indicates that GR induces chemotherapeutic resistance and results in poor prognosis in cancers [24,25]. Another GC, budesonide, also inhibits breast cancer cell proliferation and CSC formation (data not shown). Budesonide and ciclesonide are distinguished by their bulky hydrophobic groups at positions 16 and 17 [26]. Ciclesonide, dexamethasone, and prednisone show different effects on breast CSC formation. GCs regulates Hedgehog pathway activity [26]. Prednisone promotes Smoothened (SMO) accumulation and induces Hedgehog stimulation, but both ciclesonide and budesonide inhibit SMO ciliary localization and signaling activity [26]. Cyclopamine, a potent hedgehog signaling antagonist, inhibits breast cancer proliferation independent of SMO [27].

We showed via GR siRNA that ciclesonide-induced ubiquitin-dependent GR degradation and GR play an important role in CSC formation. Ciclesonide did not alter mRNA level of GR while leading the protein degradation. GR is targeted for destruction by the ubiquitin pathway. Ligand-dependent downregulation of GR has been known to limit hormone responsiveness. The GR is a phosphoprotein and becomes hyper-phosphorylated upon ligand binding. The phosphorylation of GR plays a prominent role in GR protein turnover because phosphorylation is a key signal for ubiquitination through E3 ligase. The GR antagonist RU-486 also inhibited CSC formation. Our data showed that ciclesonide inhibits breast CSCs through GR signaling regulation. GCs induce chemoresistance and tumor relapse, as evidenced by the unexpected expansion of cancers that are resistant to therapy and highly metastatic [1,10,28].

CSCs cause drug resistance, recurrence, and metastasis, which are major reasons for cancer mortality [29]. Induction of CSCs is found in a wide range of human cancers with induced YAP activation [2]. YAP acts as a major inducer of CSC formation by upregulating Sox2 and Sox9 [30,31]. The YAP protein plays a role in CSC maintenance. Ciclesonide reduced the protein level of the Hippo transducer YAP, and GR siRNA induced decreases in the YAP protein level and inhibited mammosphere formation. The YAP inhibitor verteporfin inhibited CSC formation, along with CTGF and CYR61 gene transcription. The ciclesonide-regulated GR/YAP pathway in turn regulated breast CSC formation.

Ciclesonide, a new inhaled corticosteroid, reduces airway inflammation through a single daily administration and controls asthma in atopic children [12]. Ciclesonide binds efflux transporters, p-glycoprotein, and breast cancer resistance protein (BCRP), and inhibits efflux [32]. For the first time, we showed that ciclesonide inhibits the function of the GR and YAP proteins. In addition, we showed that the GR/YAP1 pathway regulates breast CSC formation. Our results showed that the GR/YAP signaling pathway regulates breast CSC formation and revealed a new approach for targeting GR and YAP to inhibit CSC formation. Ciclesonide may extend knowledge about new concepts in anticancer therapy and inflammation.

## 5. Conclusions

In this study, we showed that ciclesonide inhibits breast cancer cell proliferation and CSC formation. Ciclesonide-induced GR degradation was dependent on ubiquitination. We showed via GR siRNA that GR plays an important role in CSC formation. The GR antagonist RU-486 also inhibited CSC formation. Ciclesonide reduced the protein level of the Hippo transducer YAP. GR siRNA induced a decrease in YAP protein expression and inhibited mammosphere formation. The YAP inhibitor verteporfin inhibited CSC formation, along with CTGF and CYR61 gene transcription. Ciclesonide inhibited the function of the GR and YAP proteins. The GR/YAP pathway regulated breast CSC formation. Thus, our results showed that the GR/YAP1 signaling pathway regulates breast CSC formation and revealed a new approach for targeting GR and YAP to inhibit CSC formation.

## Figures and Tables

**Figure 1 molecules-25-06028-f001:**
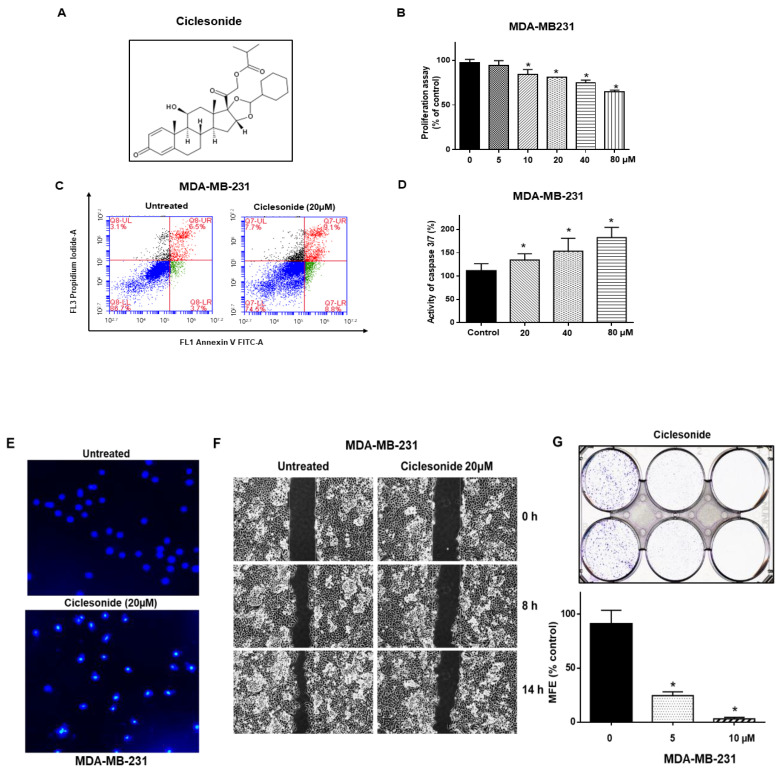
Ciclesonide reduces the proliferation of MDA-MB-231 cells. (**A**) The chemical structure of ciclesonide. (**B**) MDA-MB-231 cells were cultured in a plate with the ciclesonide. The proliferation of cancer cells was assessed with 3-(4,5-Dimethylthiazol-2-yl)-5-(3-carboxymethoxyphenyl)-2-(4-sulfophenyl)-2*H*-tetrazolium (MTS) reagent. (**C**) Ciclesonide (20 μM) induced the apoptosis of MDA-MB-231 cells. Apoptosis was assayed using Annexin V/Propidium Iodide (PI) staining. (**D**) Caspase 3/7 activity was measured with a Caspase-Glo assay kit (Promega, Madison, WI, USA). Ciclesonide increased caspase3/7 activity in a concentration-dependent manner. (**E**) Ciclesonide (20 μM) induced the formation of apoptotic bodies, as evidenced by staining with Hoechst 33342 dye (magnification, 40×). (**F**) Ciclesonide (20 μM) inhibited cell migration, as evaluated by a scratch assay. (**G**) The inhibitory effect of ciclesonide on colony formation. A total of 1000 cancer cells were cultured in plates containing 5 and 10 μM ciclesonide. Data from triplicate experiments are shown as the means ± SDs; * *p* < 0.05 vs. the control. Experiments were repeated three times.

**Figure 2 molecules-25-06028-f002:**
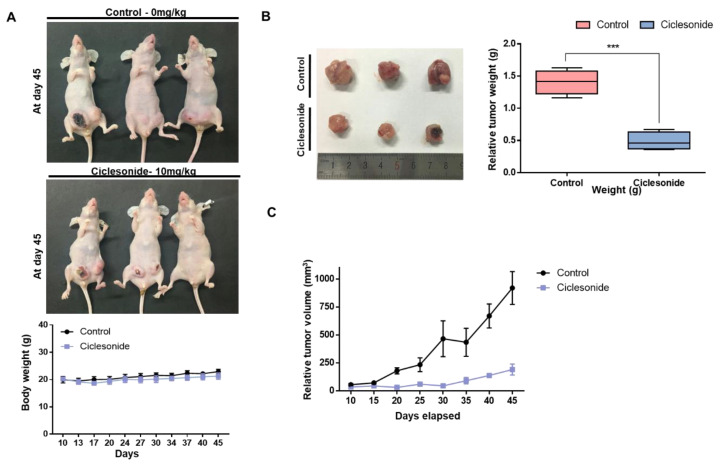
Ciclesonide reduces tumor growth in a mouse model. The 2 × 10^6^ of breast cancer cells were injected into the mammary fat pads at nude mice. The inhibitory effect of ciclesonide on tumor growth in MDA-MB-231 tumor-bearing nude mice. (**A**) The effect of tumor growth with ciclesonide in the female nude mice containing breast cancer cells. The concentration of ciclesonide was 10 mg/kg. After 45 days, images were acquired. (**B**) Inhibitory effect of ciclesonide on the tumor weight. The nude mice were sacrificed on day 45, and tumors were weighed. Data from triplicate experiments are shown as the means ± SDs; *** *p* < 0.05 vs. the control. (**C**) The volumes of tumors from nude mice during the 45-day experimental period were comparable with those of tumors from control and ciclesonide-treated mice. Tumor volumes were calculated as follows: (width^2^ × length)/2.

**Figure 3 molecules-25-06028-f003:**
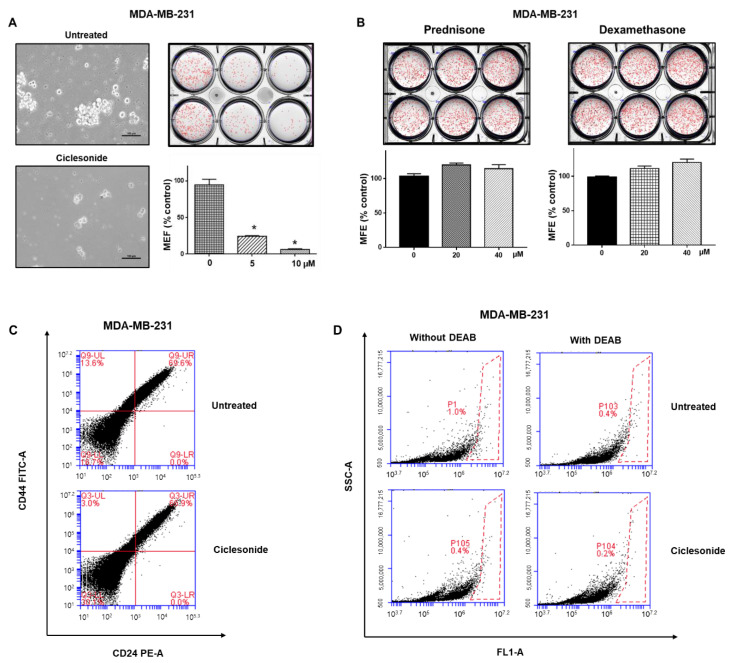
Ciclesonide reduces the formation of mammospheres from breast cancer cells. (**A**) Mammospheres were cultured for seven days in MammoCult medium. Treatment with ciclesonide (5 and 10 μM) reduced mammosphere formation to 5% that under control conditions. The experiment was conducted with cancer cells treated with ciclesonide in a single passage of spheres formation assay. * *p* < 0.05 vs. the control. Experiments were repeated three times. (**B**) Treatment with prednisone or dexamethasone (20 and 40 μM) did not reduce the mammosphere formation efficiency (MFE). The experiment was conducted with cancer cells treated with ciclesonide in a single passage of spheres formation assay. Experiments were repeated three times. (**C**) The proportion of CD44high/CD24low cells was quantified with anti-CD44-FITC and anti-CD24 PE antibodies. The CD44high/CD24low cell proportion was significantly decreased after ciclesonide treatment. (**D**) The population of ALDH1-positive cells was quantified by an ALDEFLUOR kit. The proportion of ALDH-positive cells was reduced after ciclesonide treatment. The right panel shows red-dot plots of negative control cells treated with the aldehyde dehydrogenase (ALDH) inhibitor DEAB. The left panel shows ALDH-positive cells not treated with DEAB.

**Figure 4 molecules-25-06028-f004:**
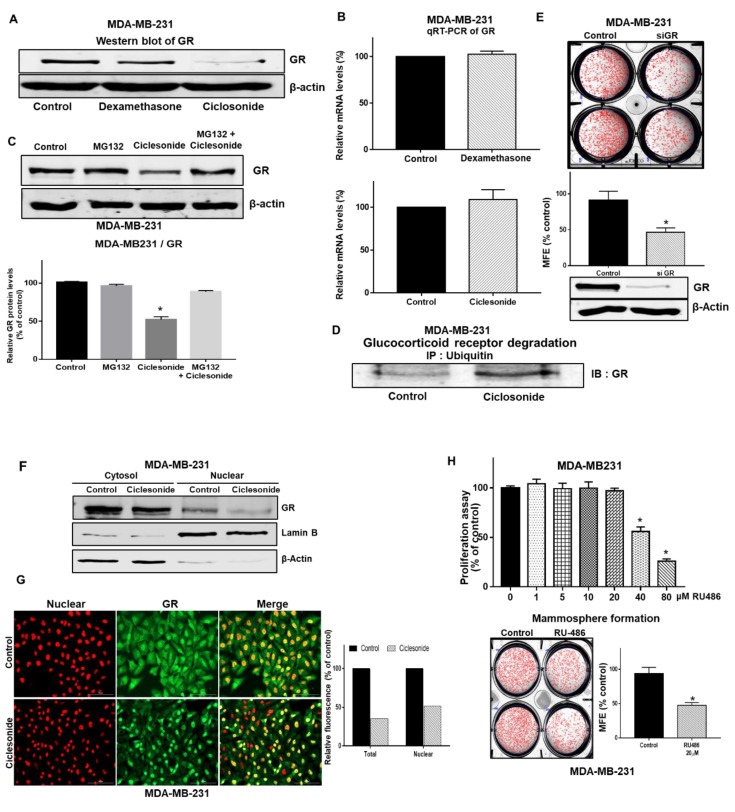
Ciclesonide blocks glucocorticoid receptor (GR) signaling pathway activation through the downregulation of GR. (**A**) Mammospheres were cultured for 5 days and incubated with ciclesonide for two days. After treatment with ciclesonide (10 μM), the expression level of GR was measured with an anti-GR antibody. Ciclesonide treatment decreased GR expression in MDA-MB-231-derived mammospheres. (**B**) After treatment with ciclesonide and dexamethasone, the transcription level of GR was assessed by real-time RT-qPCR using specific primers. We used β-actin as the internal control. (**C**) Mammospheres were cultured with MG-132 and ciclesonide for one day and lysed for immunoblot using an anti-GR antibody and β-actin as the internal control. Data from triplicate experiments are presented as the means ± SDs; * *p* < 0.05 vs. the control. Experiments were repeated three times. (**D**) Mammospheres were incubated with ciclesonide (10 μM) for 24 h and lysed for immunoprecipitation with an anti-ubiquitin antibody and immunoblot analysis with an anti-GR antibody. (**E**) The effect of GR protein on mammosphere formation was assessed using GR siRNA. Mammospheres derived from GR siRNA-treated cells were cultured for seven days in complete MammoCult medium. The experiment was conducted with cancer cells treated with ciclesonide in a single passage of spheres formation assay. The mammosphere formation efficiency (MFE) was determined. Data from triplicate experiments are shown as the means ± SDs; * *p* < 0.05 vs. the control. Experiments were repeated three times. (**F**) Mammospheres were treated with ciclesonide (10 μM) for one day. Cytosolic and nuclear proteins were separated on a 10% SDS-PAGE gel and subjected to immuno-blotting with an anti-GR antibody. β-Actin and Lamin B were used as the loading controls for the cytosolic and nuclear protein extracts, respectively. (**G**) Ciclesonide reduced the levels of cytosolic and nuclear GR protein (green) in MDA-MB-231 cells, as evidenced by immunofluorescence. Nuclei were stained with DAPI (red), and GR was labeled with an anti-GR antibody (green). Magnification, ×100. Relative fluorescence of total and nuclear GR was estimated by Gen5 cell imaging program (Appendix A). (**H**) Effect of RU486, a GR antagonist, on mammosphere formation. Cancer cells were cultured in a plate with the ciclesonide. Proliferation was evaluated with MTS reagent. Mammosphere formation was examined by determining the MFE with RU486 (scale bar = 100 μm). The experiment was conducted with cancer cells treated with ciclesonide in a single passage of spheres formation assay. Data from triplicate experiments are presented as the means ± SDs; * *p* < 0.05 vs. the control. Experiments were repeated three times.

**Figure 5 molecules-25-06028-f005:**
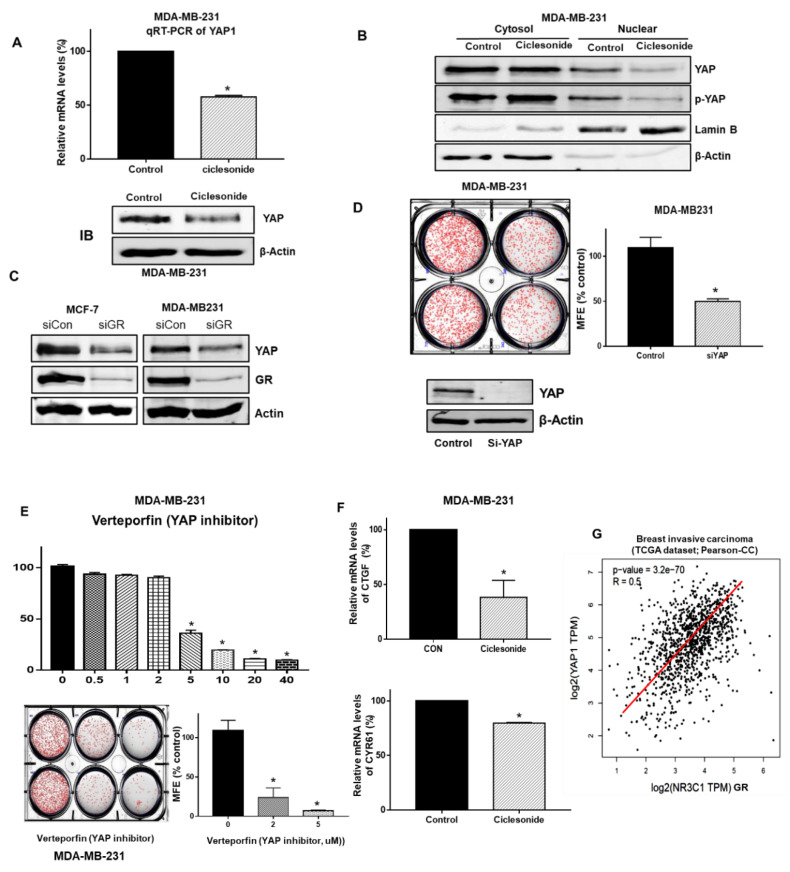
Ciclesonide reduced Yes-associated protein (YAP) nuclear localization and regulated breast CSC formation through YAP signaling. (**A**) After treatment with ciclesonide (10 μM), the transcript and protein levels of YAP1 were measured by real-time RT-qPCR using specific primers and an anti-YAP1 antibody, respectively. We used β-actin as the internal control. (**B**) Mammospheres were treated with ciclesonide (10 μM) for 1 day. After separation on 10% PAGE gels, cytosolic and nuclear protein extracts were transferred to membranes, followed by western blotting with anti-YAP and anti-p-YAP1 antibodies. β-actin and Lamin B were used as loading controls for the cytosolic and nuclear protein extracts, respectively. (**C**) siRNA-mediated silencing of GR decreased the expression of YAP and GR proteins in MCF-7 and MDA-MB-231 cells. GR protein expression was downregulated via GR siRNA, and the protein levels of YAP1 and GR were measured with anti-GR and anti-YAP1 antibodies. (**D**) siRNA-mediated silencing of YAP1 induced a significant decrease in the MFE in MDA-MB-231 cells. YAP1 was downregulated in MDA-MB-231 cells by YAP1 siRNA, and a mammosphere formation assay was performed with these cells. The experiment was conducted with cancer cells treated with ciclesonide in a single passage of spheres formation assay. (**E**) Treatment with verteporfin, a suppressor of YAP-TEAD complex formation, resulted in a decrease in the MFE in MDA-MB-231 cells. The cancer cells were cultured in a 96-well plate with the indicated concentration of verteporfin. The proliferation of the cells was measured with MTS reagent. Data from triplicate experiments are shown as the means ± SDs; * *p* < 0.05 vs. the control. Experiments were repeated three times. Mammospheres were cultured for 7 days in MammoCult medium. After treatment with verteporfin (2 and 5 μM), mammosphere formation was evaluated. The experiment was conducted with cancer cells treated with ciclesonide in a single passage of spheres formation assay. * *p* < 0.05 vs. the control. Experiments were repeated three times. (**F**) After treatment with ciclesonide, RT-qPCR of mammospheres was performed using CTGF- and CYR61-specific primers. Data from triplicate experiments are shown as the means ± SDs; * *p*
*<* 0.05 vs. the control. Experiments were repeated three times. (**G**) Correlation analysis of YAP1 and GR (NR3C1 gene) in breast cancer patients from a publicly available TCGA dataset. Analysis of the breast invasive carcinoma dataset showed that the correlation coefficient of YAP1 and GR (NR3C1 gene) was 0.5. *p*-value = 3.2 × 10^−70^.

**Figure 6 molecules-25-06028-f006:**
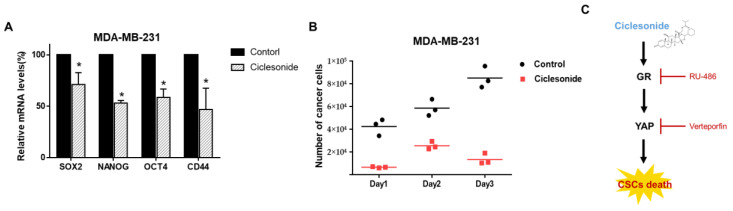
The effect of the antiasthma medication ciclesonide on CSC levels in breast cancers. (**A**) The mRNA levels of Sox2, Nanog, Oct4, and CD44 were assessed in ciclesonide-treated mammospheres using specific primers. β-actin was used as the loading control. * *p* < 0.05 vs. the control. Experiments were repeated three times. (**B**) Ciclesonide (10 μM) inhibited mammosphere growth. Ciclesonide-treated mammospheres were dissociated into single cells, and equal numbers of cells were cultured in 6 mm dishes. One day after plating, the cells were counted. After two and three days, the cells were counted in triplicate, and the mean value was plotted. (**C**) The proposed schematic suggests that ciclesonide induced GR degradation, YAP downregulation, and CSCs inhibition. GR and YAP activity contributes to CSC formation and inhibition of GR and YAP expression via siRNA suppresses CSCs formation.

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
