# Peer review of "The Antiasthma Medication Ciclesonide Suppresses Breast Cancer Stem Cells through Inhibition of the Glucocorticoid Receptor Signaling-Dependent YAP Pathway"

_molecules, 2020, doi:10.3390/molecules25246028_

Round 1

Reviewer 1 Report

In this manuscript by presented Kim, Choi, Kim and Lee, entitled “The Antiasthma Medication Ciclesonide Suppresses Breast Cancer Stem Cells through Inhibition of the Glucocorticoid Receptor Signaling-dependent YAP Pathway”, the authors demonstrate the regulatory influence of the glucocorticoid ciclesonide on the receptor as well as the Yap protein. The inhibition of both proteins correlates with reduced proliferation and migration of breast MDA MB-231 cells as well as cultured mammospheres.

The authors show some highly interesting and clinically relevant interrelations between glucocorticoids and tumor cell proliferation. Unfortunately, the data are not very coherent and the manuscript contains many correlations instead of causal relationships. One of the main questions is the use and the conclusions of so-called cancer stem cells. Based on the missing descriptions of CSC isolation, culture, and so on, I am afraid the use of mammospheres produced by (which) cells or cell line? has been ranked as cancer stem cell model. If this is true the conclusions drawn by the authors are questionable.

Main criticism:

To my opinion, a central question raised in this study is not sufficiently explained:

Ciclesonide reduces proliferation of MDA-MB231 cells as well as cultured mammospheres and in parallel reduces the stability and amount of glucocorticoide receptor (GR) protein. Unfortunately, the authors present no evidences about the responsible molecular mechanism by which ciclesonide induces GR ubiquitination. Does ciclesonide induce this effect by binding to the GR or via additional signaling pathways? Which ubiquitin ligase is involved?

This has to be answered.

The statement that “ciclesonide-treated cells showed lower levels of nuclear GR …” (lines 138/9) is not supported by the presented data in Fig. 4G.  Whereas the cytosolic localization of GR is clearly reduced in ciclesonide-treated cells, the nuclear staining is not altered very much. Please explain this obvious contradiction.

The use of so-called “mammospheres” represents a central part of the study, but the used mammospheres are insufficiently explained. The authors stated in Methods, that cancer cells were used for establishment of mammospheres. What was the origin of these cancer cells? Please give a detailed description of the used method.

Do the authors refer to these mammospheres when they describe the effects on cancer stem cells (CSC)? The origin of these CSC is not described:

Where is the origin of the CSCs?

Where is the characterization of these CSCs?

What is the difference between Fig. 3A and Fig.6B?

The conclusions drawn by the authors are often not entirely supported by the data. The description of correlated events (for instance reduction of YAP and Sox, Nanog expression) is not a conclusive evidence that YAP is the cause of the reduced SOX, Nanog expression.

Another example is the unproved prediction about the different effects of ciclesonide and dexamethasone given in the discussion in line 333 or line 342.

The manuscript has to be carefully revised regarding these overstatements.

The conclusion at the end of the chapter 2.6. is also not correct. Line 181: “The results show that ciclesonide regulates the GR/YAP signaling axis to promote breast cancer formation”. This conclusion seems to be inverted. Ciclesonide treatment resulted in reduced mammosphere formation (Fig. 1, 2, 3A,…)

The authors use the terms cell growth and cell proliferation synonymously, but this is not correct. The authors mean cell proliferation in most cases and should substitute cell growth, because the latter refer to the size of a cell.

The authors should give evidence that RU486 inhibits GR in the MDA-MB231 cell system.

The experimental setup and statistics are not clearly described. Please specify

Experiments should be performed in a least three independent experiments. One experiment done in triplicate is not enough. Please give details for every experimental assay.

Minor criticism:

Please define every abbreviation at its first used in the text. MFE, for instance, is explained quite late in Material and Methods.

Please explain the rational behind CD44+/CD24- cell populations as well as ALDH-positive cell proportions.

In Fig. 4c, the difference induced by Ciclesonide is quite small, compared with data shown in Fig4a. Please explain the possible reasons and give a densitometric quantification of the data, at least from three independent experiments.

Author Response

We submitted reviewer's comment.

Reviewer 2 Report

The paper presents the novel role of ciclesonide (a common glucocorticoid used in the treatment of asthma and allergic rhinitis) on breast cancer and cancer steam cells (CSC), showing a tumor suppressor function. They propose a mechanism identifying proteins of the Hippo pathway regulated by Ciclesonide: the GR/YAP pathway proteins regulate the BC and CSC growth.

Ciclesonide induces the glucocorticoid receptor (GR) downmodulation via proteasomal degradation through ubiquitination. The authors demonstrate the importance of GR in breast cancer and CSC growth as the siRNA for GR and the specific inhibitor RU-486 strongly inhibits CSC growth, evaluated by MEF %. Ciclesonide inhibits the protein levels of YAP, and GR downmodulation exerts the same negative effect on YAP protein underlying the signaling cascade modulating breast cancer growth.

The authors have finally demonstrated the specificity of the treatment through the administration of similar glucocorticoids (dexamethasone and prednisone) that does not show the same effects onto BC and CSC.

The experiments were conducted with well-structured approaches and reach novel interesting conclusions, supported by the analysis of the data presented. However, I have some concerns and experimental hints that should implement and support the conclusions, required for the acceptance:

  • In the Figure 1 the authors show the tumor suppressive effect of Ciclesonide on MDA-MB-231 breast cancer cells, as inhibition of proliferation and induction of apoptosis: although citated in M&M, the other breast cancer line MCF-7 is poorly shown in the experimental description. The authors should validate the MDA-MB-231 data of proliferation and apoptosis in MCF-7 cells to draw conclusion of a general BC inhibitory effect. Moreover, in this regard, a crucial point as a novel anti-cancer treatment is the assessment of possible toxicity on normal cells (in the systemic use), to endup in therapy. MCF10-A normal breast epithelial cell line should be analyzed, adding data in the current manuscript.
  • Another point discussed in Figure 1 (F) is the role of ciclesonide to inhibit BC cells migration in the wound healing assay: the experiment shown must be exerted in starving-serum free medium (a condition where cells can’t proliferate but are allowed to migrate), otherwise the data shown are a consequence of both proliferation blockade (and/or apoptosis induction) and migration, that are Eventually, 3D transwell migration assay of BC cells towards a gradient of FBS could allow a correct evaluation of the migration potential, especially for cancer cell lines defined by higher rate of proliferation.
  • Ciclesonide is shown to regulate the GR/YAP pathway, and the experiments presented are convincing in showing a GR-dependent YAP protein modulation: to verify the signaling pathway cascade proposed, the authors should insert data showing that GR levels are not affected by YAP downmodulation through siRNA and chemical inhibition (verteporfin). In other words, the authors should complement the hierarchical pathway cascade leading to tumor suppression in the paper.
  • Mammospheres assay to assess Breast CSC formation in Figure 1-3-4-5: ciclesonide reduces the formation of mammospheres from breast cancer cells. In case the experiment has been conducted with cancer cells treated with ciclesonide in a single passage of spheres formation assay, the results of the MEF% is not the outcome of a decreased CSC growth but includes the effects onto progenitors. To verify the treatment effect onto CSC numbers two or more plating passage of spheres should be performed after treatment (as in Figure 6): if this is the case please indicate the method within the figure legends, otherwise remove the statement “effect on CSC”.

Minor points:

  • Figure 1 (E): provide a magnification of the nuclei to allow the visualization of apoptotic bodies by
  • Figure 5 (B): provide WB of lamin and b-actin in both cytosolic and nuclear fraction to stress the purity of the preparations and the specificity of the signals.

Author Response

We submitted reviewer's comment.

Reviewer 3 Report

In this manuscript, the authors showed that ciclesonide, a glucocorticoid, inhibits the proliferation of breast cancer cells and CSC formation. However, similar glucocorticoids, dexamethasone and prednisone, do not inhibit CSC formation. The authors also showed that ciclesonide inhibits YAP signaling and also the expression of cancer stem cell markers. I think this study is potentially interesting. The study is well designed. This manuscript can be accepted for publication after minor revision.

Minor Revision:

  • First of all, Authors used quite similar explanation with their previous published paper in IJMS. Please rewrite the paper and pay attention for self-plagiarism issue. At same time, I believe they need to improve English spelling and grammar mistakes
  • My concern is about drug concentration used in vivo animal models. Is it comparable to the current recommended levels of drugs used in asthma treatment? How they decide the drug concentration for in vivo experiments.  
  • In introduction part line 46-49, There are five splice variants of GR. I wonder that which variants their primary antibodies can recognize in this study? Please give more detail about GR variants and the specification of primary antibodies used in this study, especially for GR protein.
  • In figure 1, dose-dependent results indicating the inhibition of breast cancer cell growth and migration. How many times of treatment was performed in dose-dependent experiment. In panel F, experiment was performed for 18h, but what about others?
  • In the firs result (Figure 1E), they mentioned that after treatment with ciclesonide, cells exhibited apoptotic body formation. How they can conclude that cells exhibit apoptotic body formation by staining with Hoechst 33342 dye. Because it is difficult to distinguish of apoptotic cell body formation by looking panel E. Please explain or discuss about apoptotic body formation by using this technique.
  • Line 87,88; authors claim that ‘the body weights of control and ciclesonide-treated mice did not change’. However, there is no quantitative data of body weights of animals. Please show your quantitative data.
  • In figure 3D, they used ALDH-positive cells under ciclesonide treatment. What is the purpose of using ALDH-positive cells? The readers may be confused in this issue. Please explain briefly in result part and discuss about it in the discussion part.
  • Ciclesonide did not alter mRNA level of GR while leading the protein degradation. Please discuss more about this issue.
  • Figure 4G, they claim that ciclesonide-treated cells showed lower levels of nuclear GR. However, this data only based on observation. Based on the pictures in panel G, GR localize both cytosol and nuclear in control group whereas GR (green) signal intensity seems like higher in nuclear with ciclesonide treatment compared to control group. Although ciclesonide inhibits nuclear GR level (based on WB data), it is hard to say GR level reduced by ciclesonide by looking IF data. Please quantitate your IF data considering co-localization analyses.

Author Response

We submitted reviewer's comment.

Round 2

Reviewer 1 Report

Review of the revised manuscript by presented Kim, Choi, Kim and Lee, entitled “The Antiasthma Medication Ciclesonide Suppresses Breast Cancer Stem Cells through Inhibition of the Glucocorticoid Receptor Signaling-dependent YAP Pathway”

I would like to acknowledge the revision of this manuscript by the authors. Unfortunately, the authors did not add significant amounts of new data to overcome the impression of a study reporting too preliminary results.

  1. ... Are evidences about the responsible molecular mechanism by which ciclesonide induces GR ubiquitination. …

Replay authors:... We want to find GR-dependent E3 ligase.

We will study E3 ligase as further study and added your comments at Discussion part as followed; ...

Comment:  The results presented here are very preliminary

  1. The statement that “ciclesonide-treated cells showed lower levels of nuclear GR …” (lines 138/9) is not supported by the presented data in Fig. 4G. Whereas the cytosolic localization of GR is clearly reduced in ciclesonide-treated cells, the nuclear staining is not altered very much. Please explain this obvious contradiction.

Replay authors: … We add a new Figure 4G containing Quantitation data and Supplementary Figure S1. We used Gen5 cell imaging program of Lionheart FX machine for quantification of GR fluorescence. First of all, we selected around 100 objects in one picture and measured the fluorescence of each object. Total and nuclear fluorescence with/without ciclesonide were determined as followed

Comment: A detailed description about the used method and algorithm has to be added to material and methods. The quantification was done on how many slides/images? What does it mean that only a few cells show nuclei without GR staining?

  1. The use of so-called “mammospheres” represents a central part of the study, but the used mammospheres are insufficiently explained. The authors stated in Methods, that cancer cells were used for establishment of mammospheres. What was the origin of these cancer cells? Please give a detailed description of the used method.

Do the authors refer to these mammospheres when they describe the effects on cancer stem cells (CSC)? The origin of these CSC is not described:

Where is the origin of the CSCs?

Where is the characterization of these CSCs?

What is the difference between Fig. 3A and Fig.6B?

Replay authors: →The reviewer’s point was well taken. We added new sentences at material method as your comments as followed;

Comment: The use of 3D cultures such as mammospheres used herein increases the informative value and reliability of cell cultures. But the denomination as cancer stem cells is misleading. Established and longley cultured cells derived from an tumor may share some characteristics with cancer stem cells but can not be equalized with CSC

  1. The conclusions drawn by the authors are often not entirely supported by the data. The description of correlated events (for instance reduction of YAP and Sox, Nanog expression) is not a conclusive evidence that YAP is the cause of the reduced SOX, Nanog expression.

Another example is the unproved prediction about the different effects of ciclesonide and dexamethasone given in the discussion in line 333 or line 342.

The manuscript has to be carefully revised regarding these overstatements.

Replay authors:

Comment: the discussion was revised

  1. The conclusion at the end of the chapter 2.6. is also not correct. Line 181: “The results show that ciclesonide regulates the GR/YAP signaling axis to promote breast cancer formation”. This conclusion seems to be inverted. Ciclesonide treatment resulted in reduced mammosphere formation (Fig. 1, 2, 3A,…)

Comment:  ok

  1. The authors use the terms cell growth and cell proliferation synonymously, but this is not correct.

Comment: ok

  1. The authors should give evidence that RU486 inhibits GR in the MDA-MB231 cell system.

Comment:   ok

  1. The experimental setup and statistics are not clearly described. Please specify

Experiments should be performed in a least three independent experiments. One experiment done in triplicate is not enough. Please give details for every experimental assay.

Replay authors:        Experiments were repeated three times.

Comment: Excuse me please, but in this case you have 4 experiments: the experiment and 3 repeats. The experiments were conducted three times.